


# Strategies to increase the accessibility of tsunami shelters enhances their adaptive capacity to risks in coastal port cities: The case of Nagoya City, Japan

Weitao Zhang[1], Jiayu Wu[2], Yingxia Yun[3]

[1] School of Architecture, Tianjin University, Tianjin, 300072, China; zhangwt2015@outlook.com

[2] College of Urban and Environmental Sciences, Peking University, China; wujiayula@gmail.com

[2] School of Architecture, Tianjin University, Tianjin, 300072, China; yunyx@126.com

*Correspondence to: Weitao Zhang(zhangwt2015@outlook.com) and Yingxia Yun (yunyx@126.com)*

**Abstract.** Coastal areas face a significant risk of tsunami after a nearby heavy earthquake. Comprehensive coastal port cities
often complicate and intensify this risk, due to the high vulnerability of their communities and liabilities associated with secondary damage. Accessibility to tsunami shelters is a key measure of adaptive capacity in response to tsunami risks and should therefore be enhanced. This study integrates the hazards that create risk into two dimensions: hazard-product risk and hazard-affected risk. Specifically, the hazard-product risk measures the hazard occurrence's probability, intensity, duration, and extension in a system. The hazard-affected risk measures the extent to which the system is affected by the hazard
occurrence. This enables the study of specific strategies for responding to each kind of risk, to enhance accessibility to tsunami shelters. Nagoya City in Japan served as the case study; the city is one of the most advanced tsunami-resilient port cities in the world. The spatial distribution of the hazard-product risk and hazard-affected risk was first visualized in 165 School District samples, covering 213 km² using a Hot Spot Analysis. The results suggest that the rules governing the distribution of these two-dimensional (2D) risks are significantly different. By refining the tsunami evacuation time-space
routes, traffic location related indicators, referring to three-scale traffic patterns with three-hierarchy traffic roads, are used as accessibility variables. Two-way Multivariate Analysis of Variance (MANOVA) was used to analyse the differences in these accessibility variables to compare the 2D risk. MANOVA was also used to assess the difference of accessibility between high-level risk and low-level risk in each risk dimension. The results show that tsunami shelter accessibility strategies, targeting hazard-product risk and hazard-affected risk, are significantly different in Nagoya. These different strategies are
needed to adapt to the risk.

## 1 Introduction

In coastal areas, a nearby maritime earthquake is typically followed by a chain of onshore waves. Some of these waves have the potential to become a heavy tsunami, significantly endangering the lives of the resident population. Comprehensive coastal port cities are places that have maintained leading positions in both global urban and port systems (Lee et al., 2008;
Cerceau et al., 2014) and both complicate and intensify risks from tsunami in the following ways. First, social-economic elements are clustered in a dense, disproportionate, and interwoven land-use pattern (Smith et al., 2011; Daamen and Vries, 2013; Bottasso et al., 2014; Ng et al. 2014; Wang et al. 2015) along lowland and flat terrain (Mahendra et al., 2011). This leads to communities having a nonlinear sensitivity to tsunami, due to the extreme diversity of hazard-affected environments. Second, port-industry land use and large disposal infrastructures are prone to fire, explosion, and chemical leakage in
response to heavy surge waves. This exposes communities to large-scale secondary damage, especially where huge port-industry complexes penetrate into residential and service zones.

To reduce population loss during a tsunami, evacuation planning (Glavovic et al., 2010; Wegscheider et al., 2011) in comprehensive coastal port cities should be consistent and supported by an effective tsunami shelter layout (Dallosso and





Domineyhowes, 2010). Tsunami shelters are typically high-rises with many floors, a large volume, and a reinforced concrete

structure. They are also generally anti-seismic, fire-proof, and explosion-proof (Scheer et al., 2012; Suppasri et al., 2013; Chocl and Butler, 2014). In most countries around the world, tsunami shelters are in public service buildings (Disaster Prevention Plan of Tokyo Port Area, 2016; Faruk et al., 2018). These buildings provide short-term emergency sheltering and long-term shelter.

Tsunami shelters are also effective shelters to protect against other surge and wind hazards caused by meteorological factors

that are intensified by climate change (Solecki et al., 2011), such as storm tides. The difference between these events is that the available time for tsunami evacuation is far shorter, generally ranging within 30 to 60 min after a heavy earthquake (Atwater et al., 2005). This makes on-site evacuation for tsunamis equally, or more important than cross-regional evacuation. (Cross-regional evacuation is the major evacuation route for other surge and wind hazards.) However, it can also lead to congested and disordered evacuation traffic to tsunami shelters during an emergency. Therefore, maximizing accessibility

from disaster areas to tsunami shelters is a key principle, especially when determining the effect of the tsunami shelter layout in the evacuation planning of coastal cities.

In general, studies on shelter accessibility have examined a broad array of factors, including traffic location assessment using simple qualitative research and quantitative evaluation (Thanvisitthpon, 2017; Rus et al., 2018; Faruk et al., 2018), facility location modeling (Ng et al., 2010; Kulshrestha et al., 2014; Mollah et al., 2017) and route optimization planning (Campos et

al., 2012; Goerigk et al., 2014; Khalid et al., 2018) using complex overall planning modeling and computer simulations. The studies specific to tsunami shelter accessibility have mainly focused on two aspects: evacuation traffic system on a technical level and adaptive capacity with an environmental focus.

Evacuation traffic systems include traffic patterns, roadways, and possible traffic congestion on roadways. With respect to traffic patterns, diverse simulations using an agent-based model have assessed evacuations to tsunami shelters on foot, in

vehicles, or using a combination of both (Mas et al. 2012). Traffic flow models (Johnstone and Lence 2014) in a disaster scenario have also been done. These simulations are applied to predict the refuge-related preferences of the population, or to recommend a tsunami evacuation traffic pattern. To guide or verify theoretical studies, social survey and questionnaire methods have been combined with statistical analyses using information from actual tsunamis that have occurred (Murakami et al., 2014).

With respect to tsunami evacuation roadways, qualitative studies have focused on roadway design, based on a city's actual road situation (Yang et al., 2010). Recommendations for modifying, enhancing, and extending city roads have also been proposed to maximize effective access to shelters across regions during tsunami evacuation (León and March, 2014). In contrast, quantitative studies are more complicated, and apply overall planning models and computer techniques. These studies have focused on establishing an evacuation network model, to identify the optimal roadways to shelters while

minimizing evacuation time and traffic cost, and maximizing the scale of evacuation (Shen et al., 2016). Furthermore, traffic congestion that extends evacuation time, including road damage and traffic accidents, have also been widely considered in evacuation network modeling and are consistently incorporated in roadway optimization planning (Chen et al., 2012; Stepanov and Smith, 2009).

In addition to tsunami shelter accessibility studies related to a specific evacuation traffic system, several studies have also

used tsunami shelter accessibility as a key measure to assess the vulnerability of coastal communities to tsunami events. Population-at-risk scales can be obtained by measuring the evacuation completion time, hazard zones, and levels (Wegscheider et al., 2011; Isabelle et al., 2016). Meanwhile, other studies have emphasized the importance of evacuation guidance, early warnings, and route planning to respond to tsunami risks (Goseberg et al.; 2014). These studies have also

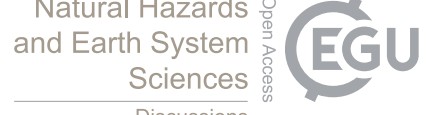



quantified and explored evacuation modeling using computer techniques, such as geographic information system (GIS)
analysis and multi-agent simulations.

To summarize existing tsunami shelter accessibility studies, they closely connect integrated hazard risks, from local to global scales. These studies are supported by multiple evacuation-space scales and multiple traffic patterns. However, although all-round traffic conditions are considered in tsunami shelter accessibility studies, few studies have correlated shelter accessibility and specific, disintegrated risk in an extended way. However, studies on hazard risk (and its evaluation) are increasingly popular (Balica et al., 2006; Yoo et al., 2011; Huang et al., 2012).

According to United Nations Office for Disaster Risk Reduction (UNISDR), hazard risk refers to the products of hazards, as well as the vulnerability of hazard-affected bodies (the system or monomer affected by hazards, such as residents, facilities, and assets) (Wamsler et al., 2013; Johnstone and Lence, 2014). Thus, hazard risk can be split into two dimensions: the hazard-product risk and the hazard-affected risk. The hazard-product risk dimension refers to the hydro-geographic system measurements of the hazard occurrence probability, intensity, duration, and extension factors (Preston et al., 2011; Goseberg et al., 2014). The hazard-affected risk dimension covers both the social-economic and political-administrative system (Felsenstein and Lichter, 2014; Jabareen, 2013). This dimension is divided into exposure, sensitivity, and adaptive capacity factors in studies on coastal area (Saxena et al., 2013) and climate change (Frazier et al., 2010) (and its driven hazards) vulnerability assessments. In a broad sense, adaptive capacity refers to different measures taken by hazard-affected bodies to mitigate, prepare, prevent, and respond to disasters, and to recover from them. As the major research object, adaptive capacity is studied independently, outside of the hazard-affected risk dimension. This is because it is inversely proportional to the negative risk-related factors listed above and the final integrated hazard risk (León and March, 2014; Desouza and Flanery, 2013; Solecki et al., 2011).

Consistent with the significantly different factors evaluated for hazard-product risk and hazard-affected risk, different spatial distributions between this two-dimensional (2D) risk can be formed. Shelter accessibility can be used as a key measure of adaptive capacity in responding to hazard risk; this accessibility can be ensured and enhanced where the hazard-product risk is large and in the case when the hazard-affected risk is high in a different and separately targeted way.

In summary, the main purpose of this study was to explore the correlation between shelter accessibility and both hazard-product risk and hazard-affected risk. The extreme complexity of the tsunami hazard risk situation in comprehensive coastal port cities makes them interesting and valuable to explore. As such, this study used the case study of Nagoya City in Japan, which is one of the most advanced tsunami-resilient port cities in the world. The goal was to investigate whether and how the tsunami shelter accessibility performance of Nagoya shows positive but different intra-city adaptive capacities to hazard-product risk and hazard-affected risk. The tsunami evacuation time-space routes reveal evacuation directions, and route orders are sorted and refined. Based on this, three traffic patterns with three-hierarchy roadways (mainly focusing on traffic location in a road system) are assessed separately to analyze shelter accessibility.

The main goals and steps of this study were: (1) to investigate the 2D spatial differentiation of risks, by estimating the spatial distribution of both hazard-product risk and hazard-affected risk, as well as of risk levels in each risk dimension; (2) to separately explore the different accessibility strategies that aim to enhance the adaptive capacity targeted to high hazard risk in two risk dimensions. This involves analyzing significant differences in tsunami shelter accessibility performance between hazard-product risk and hazard-affected risk, as well as between high-risk level and low-risk level in each risk dimension.



## 2 Study area

Nagoya City is a central and industrial city in the Greater Nagoya metropolitan area facing the Ise Bay, where the Nagoya Port is located. The topography is generally flat, with gentle hills to the east that connect to distant mountains. Three main rivers are positioned to the east of the fertile Nobi Plains, facing Ise Bay to the south. The Shonai River flows from the northeast of the city to the southwest, encircling the central area where the Horikawa River flows (see Fig. 1). There are 16 administrative regions in Nagoya City. The 12 western regions include offshore locations and estuaries, with a low and flat terrain. There is significant water network and traffic network coverage. The port-driven land use and residential land use are mixed in a high-density and diverse population distribution. In contrast, the four eastern regions are in a low-intensity and residential-development hilly area.

Nagoya City has learned lessons from the 1995 Great Hanshin-Awaji Earthquake-Tsunami and the 2011 Great East Japan Earthquake-Tsunami. This learning has contributed to significant progress in developing a tsunami-resilient city. The Nagoya City's Disaster Prevention City Development Plan (DPCDP) was issued in 2015. It is based on the scenario of a future maximum-impact earthquake-tsunami in the Nankai Trough. This Plan promoted the establishment of a Safe City (Dai, 2015). One major component of this plan was improving the tsunami shelter layout to facilitate an easy evacuation in coastal high-impact hazard surroundings.

## 3 Research design, variables, and method

### 3.1 Sample and data collection

School Districts served as the study's sample units. A "School District" is a technical term in Japan, and is defined as the most basic Disaster Prevention Community Unit by the local government. This study investigated 165 School Districts in the western 12 administrative regions of Nagoya City, which included 474 tsunami shelters (see Fig. 2). The total study area covers 213 km$^2$. Table 1 lists the data sets used for this study.

### 3.2 Accessibility variables

According to Disaster Prevention Plan documents from Nagoya City and other coastal cities in Japan, early during a heavy earthquake, most populations are encouraged to immediately evacuate to nearby seismic shelters (Dai, 2015). These seismic shelters are usually public open spaces that can protect evacuees from earthquakes and fires (Hossain, 2014; Islas and Alves, 2016; Jayakody et al., 2018). However, they do not protect evacuees from surge waves and flood damages. As such, populations evacuated to these seismic shelters should wait for tsunami warnings or evacuation orders. After receiving tsunami announcements issued by administrative departments, and based on the predicted available time to tsunami evacuate, the population will evacuate from these seismic shelters individually on foot to nearby tsunami shelters. They will also be organized by rescue authorities and sent by vehicles from the seismic shelters to nearby or remote tsunami shelters. The moves to the seismic shelter and then to tsunami shelters are the first transfer stages after an earthquake, but before a tsunami arrives. After the tsunami warning has been temporarily canceled or after a tsunami happened, a second tsunami transfer stage is activated to prevent possible or secondary tsunami damage, or to move people away from the damaged shelters. Populations in tsunami shelters at risk of flooding are organized to be continuously transferred by vehicles to inland or higher terrain.





There are also populations who have assembled in seismic shelters and who have received tsunami warnings, but who decide to return to their homes for different reasons, such as to contact family and protect private property (Murakami et al., 2014; Suppasri et al., 2012). Then, they may again decide to individually walk or drive to nearby tsunami shelters, or drive to tsunami shelters in inland and high terrain. These evacuation activities are all ordered through an emergency plan by local governments. Based on these major evacuation activities, the tsunami evacuation time-space routes were refined for this study (see Fig. 3). The study considers two tsunami transfer stages and three evacuation traffic patterns: on-site pedestrian evacuation, on-site vehicle evacuation, and cross-regional vehicle evacuation.

The three studied evacuation traffic patterns refer to tsunami shelter accessibility needs that are specific to multiple space-time routes under a tsunami risk. They can be measured along with three-hierarchy roads, and include a total of eight accessibility indicators. Each indicator is calculated based on the arithmetic mean of tsunami shelters in each School District sample (see Table 2).

On-site vehicle evacuation occurs on the main roads of the city. Previous studies on evacuation practice have shown that population-heavy occupied shelters are close to main roads and their junctions after the disaster (Allan et al., 2013). Thus, two spatial indicators are studied: a shelter's nearest distance to a main road to estimate the proximity to transportation, and a shelter's nearest distance to a junction of main roads to estimate the connectivity in a transportation network. Proximity provides a location advantage, ensuring the quick activation of traffic and a fast and organized evacuation in a local area. Connectivity supports multi-direction opportunities for local evacuations. Shelters with high connectivity may become a local evacuation hub.

Moreover, main roads in a local area are used for both vehicle and pedestrian evacuations, leading to traffic problems. Many studies have argued that the population cannot reach shelters in enough time, mostly due to road congestion (Chen et al., 2012; Campos et al., 2012). This results from flooding and destroyed roads, as well as chaos between people and vehicles, triggering traffic accidents. Thus, this study applied traffic congestion indicators: the ratio of population number to road length in each district sample, and the ratio of population number to number of road junctions in each district sample. This approach is based on the Critical Cluster Model created by Cova and Church (1997), where the possibility of congestion can be measured using the ratio of the number of people in a specific region to the overall capacity of exits across the region's boundaries. A higher ratio generally indicates traffic congestion, which triggers other traffic accidents on the road or junctions around tsunami shelters.

Cross-regional vehicle evacuation relies on regional expressways, to ensure massive, fast, and well-organized transportation. Proximity and connectivity indicators are also important. Proximity supports efficient transfers in a single-directed way to remote safer areas, because the local area was damaged by inundation and secondary disasters. Proximity also supports the acceptance of regional evacuees from offshore areas with heavy flooding. Connectivity supports multiple-directed evacuation opportunities throughout a large region. Connectivity also allows shelters to provide the regional base with comprehensive disaster-response activities.

Many studies have found that on-site pedestrian evacuations involve all city roads (main roads and branch roads). Because of the extremely high-density grid-network branch roads in Nagoya, most tsunami shelters are located on the corner of city roads. As such, there is no need to measure the proximity from shelters to city roads or junctions. We used the density of all roads and the density of junctions of all roads in each district sample to estimate the walking traffic coverage and connectivity in the vicinity of tsunami shelters.





### 3.3 Hazard risk variables

This study evaluated hazard-product risk and hazard-affected risk in Nagoya City separately. Four main hazards are associated with hazard-product risk in a tsunami scenario (see Table 3). The time-space impact of each hazard on the surrounding areas is very different. Before a tsunami, an earthquake has a relatively even impact on the local area, lowering its comparative risk (Chen et al., 2012). A tsunami results in different inundations due to complex bathymetric and topographic conditions (Xu et al., 2016), while the overall impact level decreases progressively from the open sea and river

ways toward inland. Explosions and fires are secondary hazards that occur with the earthquake and are generally intensified by the tsunami. Explosions have a concentrated point-shape impact around hazardous port-driven industry facilities, where the hazard effects decrease with the increased distance from the focal point, threatening the nearby living communities with sudden disruptions (Taveau, 2010; Christou et al., 2011). A fire hazard shows a planarly sprawling impact, due to inflammable material storage and non-fireproof construction sources.

With respect to the hazard-affected risk (see Table 3), the exposure factor refers to the extent to which a system at risk is exposed to a hazard. Day-time population density was selected as the exposure indicator in this study, because population is the hazard-affected body in a shelter accessibility analysis. The sensitivity explains how the system is susceptible to the forces and impacts associated with a hazard. We selected sensitivity indicators with environmental attributes that may impede or support population evacuation. Environmental attributes can also indicate the behavioral abilities of the distributed

population. Therefore, we measured sensitivity using both a building collapse indicator and an urban service indicator. The building collapse indicator refers to a low-quality environment and old-standard construction. It also indicates the distribution of vulnerable population, especially low-income people. It has been suggested that low-income people consistently have less ability to respond to hazards in an efficient and timely way, due to less education, less available information, lack of personal traffic tools, and poor facilities for disaster prevention (Murakami et al., 2014). The urban

service indicator can be used to represent the location of humanized facilities with barrier-free design. Furthermore, public service building complexes provide a high-density tsunami shelter area.

All study indicators were measured by summing the percentages of specific risk-effect areas (divided by the total area of each district sample), multiplied by the corresponding risk-level value of the risk map (Prasad, 2016). By Eq. (1), we calculated both the hazard-product risk and the hazard-affected risk in each district sample by adding the standardized

indicators (Yoo et al., 2011) that each contained.

$$Dimension\ index = \frac{X - Min}{Max - Min} \tag{1}$$

where X represents the value from each indicator, Min represents the minimum value, and Max represents the maximum value of the data set.

Equation (2) and Eq. (3) were assumed that all indicators contributed evenly to the final risk value (Prasad, 2016;

Kontokosta and Malik, 2018). This assumption provides significant flexibility with respect to the required input data and the practicability at a local level (Wegscheider et al., 2001). The tsunami hazard indicator was measured using the arithmetic mean of the inundation final depth indicator and inundation arrival time indicator. The sensitivity indicator was measured using the arithmetic mean of the building collapse indicator and urban service indicator.

$$Risk_{product} = ID_{tsunami} + ID_{explosion} + ID_{fire} = \frac{ID_{dep} + ID_{tim}}{2} + ID_{explosion} + ID_{fire} \tag{2}$$

$$Risk_{affected} = ID_{sen} + ID_{exp} = \frac{ID_{collapse} + ID_{service}}{2} + ID_{population} \tag{3}$$





### 3.4 Analytical techniques

First, we conducted a Hot Spot Analysis (Getis-Ord Gi*) using ArcGis 10.2 to visualize the spatial distribution of the hazard-product risk and the hazard-affected risk in Nagoya City. Hot Spot Analysis is used to identify both high-value (hot point) and low-value (cold point) spatial clustering with statistical significance. It is executed using a Fixed Distance Band to conceptualize the spatial relationships in ArcGis 10.2 software. Therefore, a Hot Spot Analysis can also visualize the spatial distribution of the risk levels separately for the hazard-product risk and hazard-affected risk.

After generating the spatial differentiation of both hazard-product risk and hazard-affected risk, we applied a two-way Multivariate Analysis of Variance (MANOVA), using SPSS 23 for the accessibility analysis. This analysis checked the significance of the effects of the two factors in a multivariate factorial experiment with a two-way layout (Zhang and Xian, 2012), and tested the null hypothesis of equal mean vectors across all considered groups (Todorov and Filzmoser, 2010). We used this to estimate the difference in tsunami shelter accessibility between 2D hazard risks, and the differences between different levels in each dimensional hazard risk.

The choice of statistical approach was based on variable measurements. First, we set eight continuous accessibility variables as dependent variables. We also converted two hazard risk variables into a single two-category variable (with a high-risk group and a low-risk group) by dividing the sample at the mean value as the independent variable. Second, we confirmed that the data for analysis satisfied the following main conditions: (1) The variance of eight dependent variables in each group should be homogeneous. If the variances are not homogeneous, a significant difference as a result of MANOVA cannot be confirmed as originating from the effect of the independent variable or from the self-different variance within each group. (2) A linear correlation should exist between dependent variables. (3) Neither univariate outliers nor multivariate outliers are found. (4) Dependent variables follow the normal distribution. (5) No multi-collinearity exists between dependent variables.

## 4 Results

### 4.1 Spatial differentiation of hazard-product risk and hazard-affected risk

Hot Spot Analysis results showed that hazard-product risk and hazard-affected risk exhibited considerably different spatial distributions in the 165 school district samples of Nagoya City. It shows that hazard-product risk is distributed in a relatively bipartite structure: offshore, it shows high risk; and inland, it shows low risk. Both these risks occur in a planarly extended way, with a smooth interface between the two risk areas (see Fig. 4-a and Fig. 4-c). It shows that the hazard-affected risk distribution can also be simplified into a bipartite structure. The high risk is along the fringe river (the Shonai River and the Tempaku River); the risk diminishes in a direction from offshore to the inland. In contrast, the low risk is along the central river (the Horikawa River), with a risk increasing as the direction moves from inland to offshore. Both risk directions extend in an axially extended way, and the interface between these two risk areas is wedge-shaped (see Fig. 4-b and Fig. 4-c).

### 4.2 Tsunami shelter accessibility performance in hazard-product risk and hazard-affected risk

The Multivariate Test in MANOVA found that the reciprocal action between hazard-product risk and hazard-affected risk is statistically insignificant (see Table 4). This indicates that in Nagoya City, the shelter accessibility in a specific area with a specific hazard-affected risk will not significantly differ between high hazard-product risk and low hazard-product risk, and vice versa. Moreover, this shows that accessibility indicators significantly differ between two groups of hazard-affected risk, but not for hazard-product risk. This, in turn, indicates that the tsunami shelter accessibility performance is significantly



different due to the spatial differentiation between building and environment quality. It also significantly differs based on population characteristics and distribution, rather than based on the presence of tsunami, explosion, and fire hazard.

The results of the Inter-Subjectivity Effect Test indicate no significant reciprocal actions between hazard-product risk and hazard-affected risk for any of eight dependent variables. This allows us to directly analyze the main effect of each dimensional hazard-risk on the dependent variables (see Table 5). Under the hazard-product risk and for cross-region vehicle evacuations, results show that the distance from shelter to expressway is only significantly shorter in high risk samples compared to low risk samples. The differences in distance to the junction of an expressway are insignificant. This indicates that the accessibility that enables fast and massive transfers in a single-directed, but not multiple-directed, way is more

advanced in high hazard-product risk areas. The insignificant difference between vehicle and pedestrian evacuation compared to on-site evacuations may indicate that on-site evacuation accessibility does not have a significant advantage in high hazard-product risk areas.

Under the hazard-affected risk, with respect to cross-region vehicle evacuation, the distances from the shelter to the expressway and from the shelter to the junction of the expressway are both significantly longer in high risk areas. This

means there are fewer advanced traffic locations for shelters in cross-region evacuation. Similarly, in an on-site vehicle evacuation, the distance to the junction of a main road in high risk areas is significant longer. Furthermore, in on-site pedestrian evacuation, the road density and road-junction density are both similar when comparing high and low risk samples. This indicates that all of the above aspects (including cross-region, on-site vehicle, and on-site pedestrian evacuation) provide a either a lower, or at least not more, advantage. However, both congestion indicators at the main-road

scale are significantly lower in high risk samples, as there is a lower possibility of congestion and other traffic accidents. This outcome indicates that on-site evacuations reduce traffic congestion in high hazard-affected risk areas.

## 5 Discussion

### 5.1 The formulation of spatial differentiation of hazard-product risk and hazard-affected risk

This study of Nagoya City suggests that in a comprehensive coastal port city with the co-development of urbanization and

industrialization, the distributions of hazard-product risk and hazard-affected risk exhibit significant differences.

For the hazard-product risk, the spatial bipartite structures with high risk offshore and low risk inland can be explained in three ways. First, high risk areas are located offshore and in a large-scale area, because they lie in the sprawling tsunami inundation range. This is because most modern coastal port cities rely on accelerated seaward land reclamation (land that is only a few meters above sea level) to escape the spatial limitations of inland areas; however, this results in further and wider

exposure to hydrologically disruptive events.

Second, the high risk extends inland, stemming from the highly developed port-driven industry along the shore of the city, broadly penetrating residential areas. Concentrated and specialized port-driven industries are consistently constructed in this area, creating imbalances in social and environmental systems. While this maximizes the advantages of harbors and low land prices, this move creates increased explosive hazards.

Third, the offshore area at the edge of the city, such as the southwest area of Nagoya, is a saturated delta alluvium deposit with a low bearing capacity. It can only be used as a natural foundation for low-rise and light-structure buildings. For example, wooden construction, a traditional but popular construction approach in Japan, spreads here to a large extent, further intensifying the fire-risk exposure of this area.



In contrast, low hazard-product risk areas are distributed inland, removed from the coast. In these areas, living and service communities are mainly developed, and are characterized by high elevations and solid soils.

For the hazard-affected risk, the spatial bipartite structure, with a high risk along a fringe river from offshore to inland, and a low risk along the central river from inland to offshore mainly manifest risks with respect to environmental attributes. Rich water networks that fragment the land and flow seaward are very typical for coastal port cities. In general, waterfronts are resource-rich and socioeconomically diverse areas. However, port cities are the interface of metropolitan and industrial areas and accommodate a broad mix of different urban developing phases (settlement, expansion, specialization, de-maritimization, redevelopment, and regionalization). This results in conflicting economic, social, and environmental values, leading to different management approaches and unbalanced resources. This conflict leads to very different practice construction conditions along river related to hazard-affected risk.

Therefore, the spatial bipartite structure of the hazard-affected risk can be explained in two ways. First, the areas along the fringe Shonai River and the Tempaku River from the Ise Bay to inland are older settlements, with lagging construction, and neglected city renewal. This is because they lie in a soft and low alluvial plain, which experiences long-term erosion due to the river system and is located at the marginalization of urban growth. The area along the Horikawa River flowing across the city center to the professional port zone accumulates dense social capital and is constructed to high standards. This is because it forms the main axis of urban development, with location-specific river shipping and landscape advancements. These shipping and landscape advancements are accompanied by expressway and rail transit for city expansion and massive commuting/logistics.

### 5.2 The strategy of tsunami shelter accessibility in hazard-product risk and hazard-affected risk

Based on the spatial differences in hazard-product risk and hazard-affected risk in comprehensive coastal port cities, we recommend different strategies to improve tsunami shelter accessibility. These strategies would enhance the adaptive capacity to each dimensional hazard risk identified by this study of Nagoya City.

In high hazard-product risk areas and given the significant possibility of heavy inundation, dense explosion, and sprawl fire, tsunami shelter accessibility is enhanced through cross-region evacuation instead of through on-site evacuation.Specifically, the difference in distance between shelter and expressway is significant, while the difference in distance to the junctions of an expressway are not significant. This is because large populations in high hazard-product risk areas should evacuate immediately before tsunamis; they then should engage in secondary evacuations after hazards inland from the shoreline area, in a single and definite direction. This makes multiple-direction accessibility over long distances less important. This advanced accessibility performance in the cross-region evacuation of Nagoya City is supported by the plan of using evacuation skeleton roads in response to a tsunami, as shown in the DPCDP (2015). These skeleton roads are either defined by existing expressways or by enhancing main roads, and show a significant single-direction attribute.

In contrast, in high hazard-affected risk areas with poor quality buildings, poor environmental conditions, vulnerable population aggregation, and the possibility of significant road damage, there is currently less capacity to develop fast and large-scale transfer mechanisms to more remote safer zones. There is a lower need and capacity to accommodate a massive number of evacuees from outer regions; however, we recommend increasing tsunami shelter accessibility through on-site evacuations, instead of cross-region evacuation.

It is useful to further elaborate upon several details related to the accessibility performance of Nagoya City specifically in high hazard-affected risk areas. Firstly, the weak tsunami shelter accessibility, based on the distance to the expressway and its junctions, may result from lower requirements for cross-region evacuation. However, it can also be explained by the fact

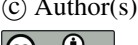



that expressways, which concentrate diverse socio-economic capital and enhance vicinity's construction quality and urban function, are underrepresented in areas highly susceptible to hazards. Second, the significantly longer distance to the main road junctions in high risk areas can also be explained by the lagging traffic; this is because main roads cannot form a perfect network. Third, the insignificant difference between road density and junction density may originate from the fact that many modern and old communities in Japan have been constructed using a similarly scaled grid-network branch-road systems.

Above all, the less or lack of advantage in cross-regional and on-site evacuation in high hazard-affected risk areas originate from the reality-based situation in lagging areas of urban construction and development. They cannot be improved in the short term. However, the lower possibility of congestion and other traffic accidents can compensate for these disadvantages. This increases the adaptive capacity of tsunami shelters, by reducing the traffic risk associated with vulnerable population activity. It also effectively limits roadblocks. This adds to the success of the land readjustment project in Nagoya City. This project is designed to develop sound city areas out of undeveloped urban areas or areas scheduled for urbanization. This development is done by exchanging and transforming disorderly land into public roads and public service facilities. This project should be continued in the long term to improve the main road network and to extend to branch roads in high hazard-affected risk areas. This would improve tsunami shelter accessibility within safe traffic environments and walkable distances. This is particularly needed because of the low ability of the vulnerable population to evacuate over long distances during an emergency and in poor building environments.

**5.3 Limitations and suggestions for future research**

This study sorted and refined the population evacuation time-space routes for accessing tsunami shelters. It did not, however, study the rescue and logistical transportation provided by the governmental and public sectors. Future studies could explore multiple transportation activities involved in accessing tsunami shelters. This would provide recommendations related to accessibility that could be used by multiple stakeholders.

Moreover, the accessibility of the studied shelters is specific to the traffic location. This relates to both traffic tools and range; but does not consider the populations accommodated by the shelters, the traffic cost, and evacuation time. Future studies should apply both overall planning modeling and computer simulations to design shelter accessibility multi-object optimization separately for high hazard-product risk areas and high hazard-affected risk areas. Those investigations could use the targeted accessibility strategies proposed in this study to as guidance.

Finally, the indicators that were used to evaluate the location-specific hazard risks in Nagoya City form a simple evaluation system due to data shortages; however, the indicators used can still reflect the main aspects of hazard risk associated with evacuations in a tsunami scenario. Future studies could expand on these indicators to produce more detailed hazard-risk information in a multiple evaluation system with a reasonable weight assessment. This could include a sub-system of disaster process phases and disaster affected bodies to achieve a more accurate hazard risk evaluation. This exploration should be closely related to a specific evacuation or relief processes.

**6 Conclusions**

This Nagoya City case study shows that in comprehensive coastal port cities, the hazard-product risk and hazard-affected risk exhibit considerably different spatial distribution rules. Here, the hazard-product risk is distributed in a bipartite structure with high risk offshore and low risk inland. Both are manifested in a planarly extended way with a smooth interface. The hazard-affected risk is distributed in a different bipartite structure, with high risk along a fringe river from offshore to



inland and low risk along a central river from inland to offshore. Both occur in an axially extended way and with a wedge-shaped interface.

Based on the spatial differentiation of 2D hazard risk, we recommend different strategies for improving tsunami shelter accessibility, with the goal of enhancing adaptive capacity for each dimensional hazard risk. This Nagoya City case study shows that in high hazard-product risk areas, tsunami shelter accessibility is enhanced by cross-region evacuations, by

increasing the proximity from shelters to regional expressways. This is a better alternative than requiring on-site evacuation. In contrast, in hazard-affected risk areas, it is recommend increasing tsunami shelter accessibility through on-site evacuations, while reducing the possibility of traffic congestion on city main roads and main road junctions. This is a better alternative than intensifying a cross-region evacuation. Tsunami shelter accessibility performance was positive with respect to the targeted adaptive capacity for different dimensional hazard risks in Nagoya. This may also provide indicators for other

coastal port cities.

**Acknowledgments:** This study is funded by the Major Project of China National Social Science Fund (2013): Study on Comprehensive Disaster Prevention Measure and Safety Strategy of Coastal City Based on Intelligent Technology (13&ZD162). The authors thank the anonymous reviewers of this essay and the participants of the Study on Comprehensive Disaster Prevention Measure and Safety Strategy of Coastal City Based on Intelligent Technology for ideas, input, and

review.

**Author Contribution:** Weitao Zhang and Jiayu Wu conceived the experiments. Weitao Zhang designed the experiments; performed the experiments; analyzed the data; and wrote the paper. Yingxia Yun guided the article structure.

**Competing Interests:** The authors declare that they have no conflict of interest.

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





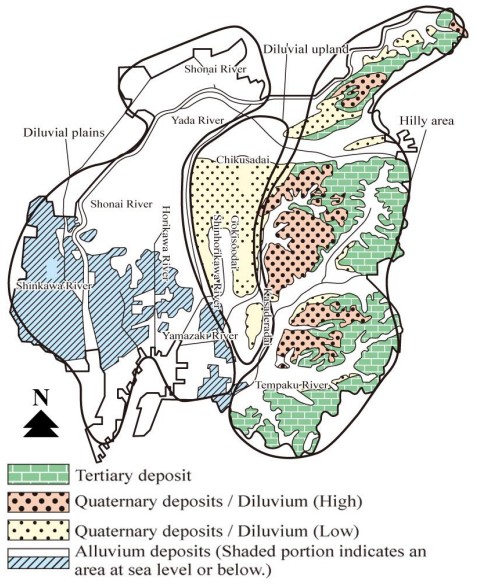


**Figure 1: The geological features of Nagoya City (Introduction of Outline Section of Planning for Nagoya, 2012. Accessed from the official website of Nagoya City: http://www.city.nagoya.jp/jutakutoshi/page/0000045893.html).**

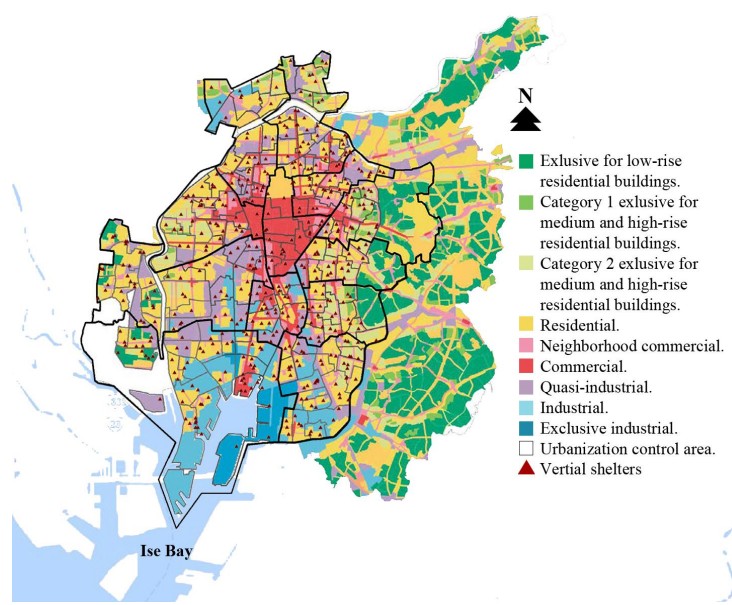

**Figure 2: The School Districts and tsunami shelters of the studied administrative regions in Nagoya City (The authors combined**

**the Nagoya City land use map from Initiatives in Planning for Nagoya, 2012. Accessed at the official website of Nagoya City: http://www.city.nagoya.jp/jutakutoshi/page/0000045893.html).**



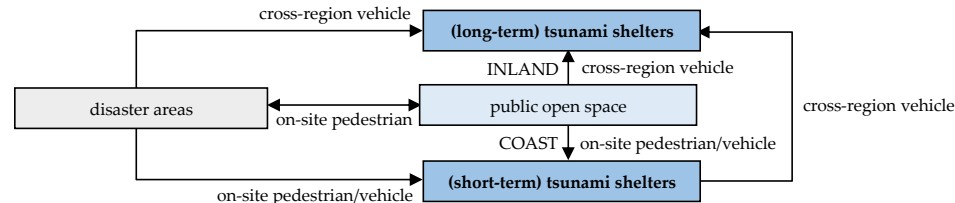

**Figure 3: Tsunami evacuation time-space routes with major evacuation activities.**

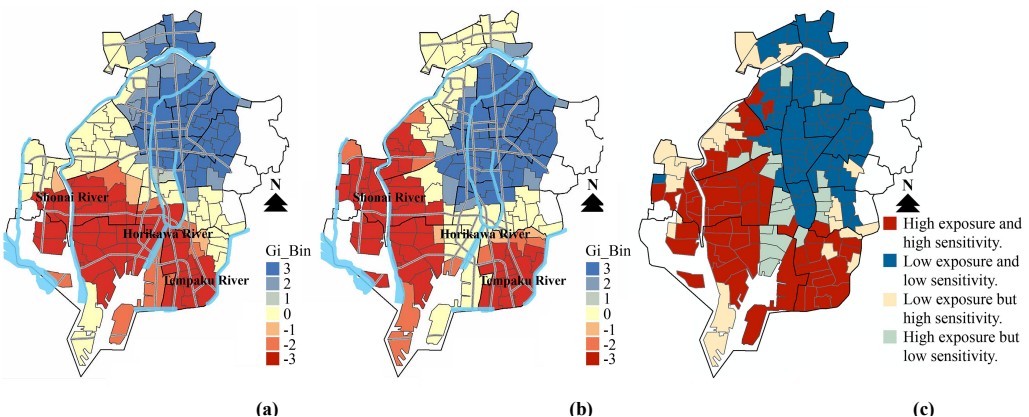

(a)                                        (b)                                        (c)

**Figure 4: Distribution of hazard risk: (a) Hot Spot Analysis outcome of hazard-product risk; (b) Hot Spot Analysis outcome of hazard-affected risk; (c) Distribution of high risk and low risk in hazard-product risk and hazard-affected risk.**



**Table 1: List of data sets used for this study.**

| Data set | Remark | Resource | Period |
|---|---|---|---|
| Tsunami inundation map | Maximum possible situation | Tsunami disaster risk assessment in DPCDP | 2015 |
| Building collapse map | | Earthquake disaster risk assessment in DPCDP | |
| Fire sprawl map | | Fire disaster risk assessment in DPCDP | |
| Land use map | Updated plan | Initiatives in planning for Nagoya | 2012/2013 |
| Shelter map | | Shelter definition and layout in DPCDP. | 2015 |
| | Current situation | Government official website | 2018 |
| | | Initiatives in planning for Nagoya | 2012/2013 |
| Road system map | Updated plan | Disaster-relief roads definition and layout in DPCDP. | 2015 |
| | Current situation | OSM website | 2018 |
| Population data | Estimated in this study | Land use map (2012); planning for Nagoya outline (2013); statistical sketch of Nagoya (2015) | -- |

**Table 2: List of accessibility indicators for this study.**

| Spatial scale | Traffic tool | Roadway | Indicator | Type | Measure |
|---|---|---|---|---|---|
| On-site evacuation | Vehicle | Regional expressway | 1 Dist_e | Length | Proximity |
| | | | 2 Dist_junc_e | | Connectivity |
| | | | 3 Dist_m | | Proximity |
| Cross-region evacuation | Vehicle | City main road | 4 Dist_junc_m | | Connectivity |
| | | | 5 C_m | Ratio | Congestion |
| | | | 6 C_junc_m | | |
| | Pedestrian | All city road | 7 Dens_a | | Proximity |
| | | | 8 Dens_junc_a | | Connectivity |

**Table 3: List of indicators computed in the hazard risk evaluation**

| Dimension | Indicator | | Risk level description (level) | Data situation |
|---|---|---|---|---|
| Hazard product | 1 Tsunami hazard | Inundation final depth | 1-6: Inundation map from 3 m-inundation area (6) to 0.3 m-inundation area (1). | Existing |
| | | Inundation arrival time | 1-6: Inundation map from 120 min-inundation area (6) to 720 min-inundation area (1). | |
| | 2 Explosion hazard | | Intensity value: 1-4: Buffers of port-industry land or critical disposal facilities (4) by 250 m (3), 500 m (2), and 1000 m (1). Hazard possibility coefficient: quasi port industry (50%), common port industry land (80%), exclusive port industry land (100%), and disposal facilities (100%) (Multiply the possibility coefficient by the intensity value.) | Mapped in this paper |
| | 3 Fire hazard | | 1-4: Fire sprawl map from 1000-building area (4) to 100-building area (1). | Existing |
| Hazard affected | 1 Sensitivity | Building collapse | 1-5: Building collapse map from 20%-building area (4) to 5%-building area (1). | Existing |
| | | Urban service | 1-4: Buffers of public service area by 250m (1), 500m (2), 1000m (3), and over 1000m (4). Hazard possibility coefficient: central service land (100%) and adjacent service land (50%) (Multiply the possibility coefficient by the intensity value.) | Mapped in this paper |
| | 2 Exposure | Population density | Day-time population. | |

**Table 4: List of Multivariate Tests.**

| Effect | | Value | F | Suppose df | Error df | Sig. |
|---|---|---|---|---|---|---|
| Hazard-product risk | | 0.910 | 1.501 | 10.000 | 152.000 | 0.144 |
| Hazard-affected risk | Wilks_Lambda | 0.764 | 4.683 | 10.000 | 152.000 | 0.000 |
| Hazard-product risk * hazard-affected risk | | 0.950 | 0.795 | 10.000 | 152.000 | 0.633 |

* The mean difference is significant at the 0.05 level.





**Table 5: List of paired comparisons.**

| Dimension | Dependent variable | Low risk | High risk | Mean value difference (I-J) | Standard error | Sig. | 95% confidence interval of difference | |
|---|---|---|---|---|---|---|---|---|
| | | | | | | | Lower limit | Upper limit |
| Hazard-product risk | Dist_e | 1 | 2 | 2.289* | 0.926 | 0.015 | 0.460 | 4.118 |
| | Dist_junc_e | 1 | 2 | 1.449 | 1.504 | 0.337 | -1.522 | 4.419 |
| | Dist_m | 1 | 2 | -0.029 | 0.120 | 0.808 | -0.266 | 0.208 |
| | Dist_junc_m | 1 | 2 | -0.113 | 0.210 | 0.591 | -0.529 | 0.302 |
| | C_m | 1 | 2 | 22.329 | 22.185 | 0.316 | -21.483 | 66.140 |
| | C_junc_m | 1 | 2 | 1.307 | 3.133 | 0.677 | -4.880 | 7.494 |
| | Dens_c | 1 | 2 | 0.028 | 0.094 | 0.762 | -0.157 | 0.214 |
| | Dens_junc_c | 1 | 2 | -0.057 | 0.072 | 0.430 | -0.199 | 0.085 |
| Hazard-affected risk | Dist_e | 1 | 2 | -3.500* | 0.926 | 0.000 | -5.329 | -1.671 |
| | Dist_junc_e | 1 | 2 | -5.287* | 1.504 | 0.001 | -8.257 | -2.317 |
| | Dist_m | 1 | 2 | -0.108 | 0.120 | 0.368 | -0.345 | 0.129 |
| | Dist_junc_m | 1 | 2 | -.0534* | 0.210 | 0.012 | -0.949 | -0.119 |
| | C_m | 1 | 2 | 91.709* | 22.185 | 0.000 | 47.898 | 135.520 |
| | C_junc_m | 1 | 2 | 9.637* | 3.133 | 0.002 | 3.450 | 15.824 |
| | Dens_c | 1 | 2 | 0.059 | 0.094 | 0.528 | -0.126 | 0.245 |
| | Dens_junc_c | 1 | 2 | 0.140 | 0.072 | 0.053 | -0.002 | 0.282 |

* The mean difference is significant at the 0.05 level.