# Peer review of "Strategies for increasing tsunami shelter accessibility to enhance hazard risk adaptive capacity in coastal port cities: A Study of Nagoya City, Japan"

_Natural Hazards and Earth System Sciences, 2018_

## Referee Comment (RC1) · Anonymous Referee #1 · 10 Dec 2018

I think study on strategies to increase the accessibility of tsunami shelters enhances their adaptive capacity to the hazard-product risk and hazard-affected risk separately is a very interesting and enlightening research. This paper has been well-written; however, there are some major concerns as follows: 1. In the title: Please reduce words number for title to make the key objective be highlighted. There is too much information in the title which makes it difficult to identify the primary objective for this study. 2. In the introduction part: The relationship among hazard-product risk, hazard-affected risk, and adaptive capacity is not expressed very clearly. Please rearrange the explanation

of their relationship. 3. In Section 3.2.: Please explain the "the first transfer stage" and "the second tsunami transfer stage" in Figure 3. 4. In Section 3.2.: Please explain "(long-term) tsunami shelters" and "(short-term) tsunami shelters" shown in Figure 3. 5. In Section 3.3.: Why "the urban service indicator can be used to represent the location of humanized facilities with barrier-free design"? More explanation is suggested to be added. 6. In Section 3.3.: Why Equation (2) and Eq. (3) were assumed that all indicators contributed evenly to the final risk value? More explanation is suggested to be added. 7. In Section 3.4.:More explanation why you use hot spot analysis to identify spatial clustering of the integrated values is suggested. 8. In Section 4.1.: How is Figure 4-c mapped? You should explain it in the text.

---

## Short Comment (SC1) · 10 Dec 2018

The research topic of accessibility of tsunami shelters adapting to hazard-product risk and hazard-affected risk separately is very interesting. However, the Nagoya City's situation is very particular. Can you talk more about how the suggestions from Nagoya guide the urban planning in practice?

---

## Author Comment (AC1) · 15 Dec 2018

Thank you for your question about the paper. Nagoya City is a very specific and inspiring case study. It effectively represents cities that have a high level of agglomeration, but have different distributions of both hazard-product factors (elevation, soil, river, dangerous source...) and hazard-affected factors (population, socio-economic capital... ). This complex situation presents more challenges for these types of cities, making insufficient safety practices a major problem. City repair and renewal is needed to address problems in this type of complex city environment.

[Figure]

This paper discusses the spatial[Did you mean spatial? Or special? ] relationship between tsunami shelters and road systems in different hazard-risk situations. The recommended accessibility-related strategies can be applied to inform both city repair and renewal planning and practice. These strategies match two core elements of city repair and renewal: the city's response to the high agglomeration of hazard-risk factors using meso/micro-strategies, and the city's response to different distributions of hazard-risk factors using a classification strategy.

First, because most of the city construction is already completed and saturated, the recommended accessibility-related strategies do not relate to city-level macro-structural adjustment, or large-scale demolition and reconstruction. I[Very long sentence; recommend splitting in two. ]nstead, they focus on shelter and road reorganization and improvement. Second, the strategies are proposed separately for the hazard-product and for the hazard-affected environment, as these are very different from each other.

In summary, this study's results and conclusion significantly contribute to practices associated with city repair and renewal planning.

---

## Author Comment (AC3) · 18 Dec 2018

1.In the title: Please reduce words number for title to make the key objective be highlighted. There is too much information in the title which makes it difficult to identify the primary objective for this study.

Reply: Thank your for your precious advice. We have reduced words number for title. The final title is "Strategies for increasing tsunami shelter accessibility to enhance hazard risk adaptive capacity: A Study of Nagoya City, Japan". "tsunami shelter", "accessibility", and "hazard risk adaptive capacity" are highlighted.

2.In the introduction part: The relationship among hazard-product risk, hazard-affected risk, and adaptive capacity is not expressed very clearly. Please rearrange the explanation of their relationship.

Reply: Thank your for your valuable suggestion. We have rearranged this paragraph as "...Therefore, in a broad sense, the hazard-affected risk is an comprehensive concept. Within this concept, exposure and sensitivity are factors that are proportional to the hazard-affected risk and the final integrated hazard risk. In contrast, adaptive capacity refers to different measures taken by hazard-affected bodies to mitigate, prepare, prevent, and respond to disasters, and to recover from them (León and March, 2014; Desouza and Flanery, 2013; Solecki et al., 2011). However, when focusing on the narrow sense of hazard-affected risk related to negative risk-related factors, adaptive capacity becomes the major research object and can be studied independently, outside of the hazard-affected risk dimension."

3.In Section 3.2.: Please explain the "the first transfer stage" and "the second tsunami transfer stage" in Figure 3.

Reply: Please forgive us for not adding them in Figure 3 and allow us to make the following explanation: When we resorted the paragraph including these two transfer notions, we decided not to add them in Figure 3. It is because the transfer order proposed is only used to express the complexity of transfer activity, but is not the point of this research. Add them in Figure 3 will make Figure 3 too complicated to make the more important objectives(sheltering and traffic) unclear. At the same time, we rewrote the sentence including "the first transfer stage" and "the second tsunami transfer stage" to weaken their role in this paper.

4.In Section 3.2.: Please explain "(long-term) tsunami shelters" and "(short-term) tsunami shelters" shown in Figure 3.

Reply: Thank your for your valuable suggestion. We have explained them in the text as "In general, only tsunami shelters in inland and high terrain can support long-term safe sheltering. In contrast, shelters in flooding-risk areas are appropriate for short-term emergency sheltering. Therefore, populations in these short-term shelters are organized in a way that allow them to be continuously transferred by vehicles to inland or higher terrain."

5.In Section 3.3.: Why "the urban service indicator can be used to represent the location of humanized facilities with barrier-free design"? More explanation is suggested to be added.

Reply: Thank your for your valuable suggestion. We have added explanation as "...That is because openness, fairness, and security are key factors in public service use. Furthermore, public service building complexes provide a high-density tsunami shelter area."

6.In Section 3.3.: Why Equation (2) and Eq. (3) were assumed that all indicators contributed evenly to the final risk value? More explanation is suggested to be added.

Reply: Thank your for your valuable suggestion. We have added more explanation as "Equations (2) and Eq. (3) assume that all indicators contributed evenly to the final risk value. This assumption provides significant flexibility with respect to the required input data and the practicability at a local level. An assumption of equal weight is preferred because of its easy comprehensively, replicability, and calculability (Prasad, 2016; Kontokosta and Malik, 2018)."

7.In Section 3.4.:More explanation why you use hot spot analysis to identify spatial clustering of the integrated values is suggested.

Reply: Thank your for your valuable suggestion. "The analysis outcome shows that high-value areas are surrounded by high-values. The reverse is also true: low-value areas are surrounded by low-values." Therefore, a Hot Spot Analysis can visualize the

spatial distribution of the risk levels separately for the hazard-product risk and hazard-affected risk. And we also added it in the text.

In Section 4.1.: How is Figure 4-c mapped? You should explain it in the text.

Reply: Thank your for your valuable suggestion. We have explained it in its capital as " (c) High exposure/sensitivity and low exposure/sensitivity are divided by the average of exposure/sensitivity value."

Please also note the supplement to this comment:
https://www.nat-hazards-earth-syst-sci-discuss.net/nhess-2018-267/nhess-2018-267-AC3-supplement.zip

―――――――――――――――――――

---

## Referee Comment (RC2) · Anonymous Referee #1 · 8 Jan 2019

Thanks for authors' responses to my comments. All my concerns have been successfully addressed in this improved version, I recommend to accept the paper in the current version.

---

## Referee Comment (RC3) · Anonymous Referee #2 · 28 Jan 2019

This study on strategies to increase the accessibility of tsunami shelters, the case of Nagoya city raises very important disaster management issues considering various access conditions. There are several points which may not be understood well and some additional explanation or revisions are advised.

Figs. 1, 2, and 4: Scales are necessary in the maps.

Table 2: As for spatial scale, the upper half lines seem to be Cross-region evacuation, and the lower half lines seem to correspond to on-site evacuation.

In Section 5.2, or in any earlier section, there should be some references and comments on the range of time after the earthquake occurrence tsunami is expected to hit along the coastal area of Nagoya city and also on the distance pedestrian evacuees or evacuees using vehicles may possibly move to and reach some safe shelters.

---

## Author Comment (AC4) · 29 Jan 2019

RC: Figs. 1, 2, and 4: Scales are necessary in the maps.

Reply: Thank you for your precious reminder. We have added the scale in the maps. The revised Figs are shown in the attachment.

RC: Table 2: As for spatial scale, the upper half lines seem to be Cross-region evacuation, and the lower half lines seem to correspond to on-site evacuation.

[Figure]

Reply: We so appreciate that you pointed out this mistake. We are sorry about it and have revised it which is shown in the attachment. Thank you for your careful correction again.

RC: In Section 5.2, or in any earlier section, there should be some references and comments on the range of time after the earthquake occurrence tsunami is expected to hit along the coastal area of Nagoya city and also on the distance pedestrian evacuees or evacuees using vehicles may possibly move to and reach some safe shelters.

Reply: Thank you for your valuable suggestions. Yes, we have obtained effective data about tsunami's arrival time and evacuation's distance which are applicable to Nagoya City study. These data are achieved mainly from official Disaster Prevention Planning documents in Japan, and reports and studies on Great East Japan Earthquake. We have added these references in Section 3.2 where we think it would be more appropriate. We also enriched Fig.3 to show these references more clearly.

Section 3.2 has been revised (in blue in the attachment) as:

"...According to Japan's coastal city disaster prevention document and Great East Japan Earthquake's experience (Disaster Prevention Plan of Tokyo Port Area, 2016; Tanaka, 2017), tsunami warnings are issued by administrative departments in 2 to 3 min after a heavy earthquake. And the tsunami will finally arrive at 30 to 60 min (Atwater et al., 2005)..." "....Combined with a statistical research on evacuation traffic patterns in Great East Japan Earthquake (H. Murakami et al., 2014), three evacuation traffic patterns can be sorted out in the multiple tsunami evacuation time-space routes: on-site pedestrian evacuation (90% pedestrian evacuation in 1000m), on-site vehicle evacuation (80% vehicle evacuation in 2000m), and cross-regional vehicle evacuation (20% vehicle evacuation over 2000m). Therefore, these three traffic patterns refer to tsunami shelter accessibility needs and are studied in this paper. They can be measured along with three-hierarchy roads, and include a total of eight accessibility indicators. Each indicator is calculated based on the arithmetic mean of tsunami

shelters in each School District sample (see Table 2)..."

Please also note the supplement to this comment:
https://www.nat-hazards-earth-syst-sci-discuss.net/nhess-2018-267/nhess-2018-267-AC4-supplement.pdf
* * *
[Figure]

[Figure]

**Fig. 1.** The geological features of Nagoya City.

The map legend contains the following entries:

- Exlusive for low-rise residential buildings.
- Category 1 exlusive for medium and high-rise residential buildings.
- Category 2 exlusive for medium and high-rise residential buildings.
- Residential.
- Neighborhood commercial.
- Commercial.
- Quasi-industrial.
- Industrial.
- Exclusive industrial.
- Urbanization control area.
- ▲ Vertial shelters

N

5km
3mi

**Ise Bay**

**Fig. 2.** The School Districts and tsunami shelters of the studied administrative regions in Nagoya City.

[Figure]

**Fig. 3.** Distribution of hazard risk.

>2000m
cross-region vehicle

earthquake → disaster areas

<1000m
on-site pedestrian

**(long-term) tsunami shelters**
INLAND
cross-region vehicle,>2000m

tsunami warnings

2-3 min

public open space

2-3 min

tsunami warnings

COAST
on-site pedestrian/vehicle,<2000m

<2000m
on-site pedestrian/vehicle

**(short-term) tsunami shelters**

canceled tsunami warnings
/ tsunami ended

>2000m
cross-region vehicle

**Fig. 4.** Tsunami evacuation time-space routes with major evacuation activities.

**Supplement:**

**RC: Figs. 1, 2, and 4: Scales are necessary in the maps.**

Reply: Thank you for your precious reminder. We have added the scale in the maps. The revised Figs are shown in the attachment.

[Figure]

Figure 1: The geological features of Nagoya City (Introduction of Outline Section of Planning for Nagoya, 2012. Accessed from the official website of Nagoya City: http://www.city.nagoya.jp/jutakutoshi/page/0000045893.html).

[Figure]

Figure 2: The School Districts and tsunami shelters of the studied administrative regions in Nagoya City (The authors combined the Nagoya City land use map from Initiatives in Planning for Nagoya, 2012. Accessed at the official website of Nagoya City: http://www.city.nagoya.jp/jutakutoshi/page/0000045893.html).

[Figure]

Figure 4: Distribution of hazard risk: (a) Hot Spot Analysis outcome of hazard-product risk; (b) Hot Spot Analysis outcome of hazard-affected risk; (c) High hazard-product/hazard-affected risk and low hazard-product/hazard-affected are divided by the average of hazard-product/hazard-affected value.

**RC: Table 2: As for spatial scale, the upper half lines seem to be Cross-region evacuation, and the lower half lines seem to correspond to on-site evacuation.**

Reply: We so appreciate that you pointed out this mistake. We are sorry about it and have revised it which is shown in the attachment. Thank you for your careful correction again.

Table 2: List of accessibility indicators for this study.

| Spatial scale | Traffic tool | Roadway | Indicator | | Type | Measure |
|---|---|---|---|---|---|---|
| Cross-region evacuation | Vehicle | Regional expressway | 1 | Dist_e | Length | Proximity |
| | | | 2 | Dist_junc_e | | Connectivity |
| | | | 3 | Dist_m | | Proximity |
| On-site evacuation | Vehicle | City main road | 4 | Dist_junc_m | | Connectivity |
| | | | 5 | C_m | Ratio | Congestion |
| | | | 6 | C_junc_m | | |
| | Pedestrian | All city road | 7 | Dens_a | | Proximity |
| | | | 8 | Dens_junc_a | | Connectivity |

**RC: In Section 5.2, or in any earlier section, there should be some references and comments on the range of time after the earthquake occurrence tsunami is expected to hit along the coastal area of Nagoya city and also on the distance pedestrian evacuees or evacuees using vehicles may possibly move to and reach some safe shelters.**

Reply: Thank you for your valuable suggestions. Yes, we have obtained effective data about tsunami's arrival time and evacuation's distance which are applicable to Nagoya City study. These data are achieved mainly from official Disaster Prevention Planning documents in Japan, and reports and studies on Great East Japan Earthquake. We have added these references in Section 3.2 where we think it would be more appropriate. We also enriched Fig.3 to show these references more clearly.

Section 3.2 has been revised (in blue) as:

"According to Disaster Prevention Plan documents from Nagoya City and other coastal cities in Japan, early during a heavy earthquake, most populations are encouraged to immediately evacuate to nearby seismic shelters (Dai, 2015). These seismic shelters are usually public open spaces that can protect evacuees from earthquakes and fires (Hossain, 2014; Islas and Alves, 2016; Jayakody et al., 2018). However, they do not protect evacuees from surge waves and flood damages. As such, populations evacuated to these seismic shelters should wait for tsunami warnings or evacuation orders.

According to Japan's coastal city disaster prevention document and Great East Japan Earthquake's experience (Disaster Prevention Plan of Tokyo Port Area, 2016; Tanaka, 2017), tsunami warnings are issued by administrative departments in 2 to 3 min after a heavy earthquake. And the tsunami will finally arrive at 30 to 60 min (Atwater et al., 2005). After receiving tsunami announcements, and based on the predicted available time for tsunami evacuation, the population will evacuate from these seismic shelters individually on foot to nearby tsunami shelters. They will also be organized by rescue authorities and sent by vehicles from the seismic shelters to nearby or remote tsunami shelters. The moves to the seismic shelter and then to tsunami shelters are the first transfer stage after an earthquake, but before a tsunami arrives.

After the tsunami warning has been temporarily canceled or after a tsunami happened, a second tsunami transfer stage is activated to prevent possible or secondary tsunami damage, or to move people away from the damaged shelters. In general, only tsunami shelters in inland and high terrain can support long-term safe sheltering. In contrast, shelters in flooding-risk areas are appropriate for short-term emergency sheltering. Therefore, populations in these short-term shelters are organized in a way that allow them to be continuously transferred by vehicles to inland or higher terrain.

There are also populations who have assembled in seismic shelters and who have received tsunami warnings, but who decide to return to their homes for different reasons, such as to contact family and protect private property (Murakami et al., 2014; Suppasri et al., 2012). Then, they may again decide to individually walk or drive to nearby tsunami shelters, or drive to tsunami shelters in inland and high terrain. These evacuation activities are all ordered through an emergency plan by local governments. Based on these major evacuation activities, the multiple tsunami evacuation time-space routes were refined for this study (see Fig. 3).

Combined with a statistical research on evacuation traffic patterns in Great East Japan Earthquake (H. Murakami et al., 2014), three evacuation traffic patterns can be sorted out in the multiple tsunami evacuation time-space routes: on-site pedestrian evacuation (90% pedestrian evacuation in 1000m), on-site vehicle evacuation (80% vehicle evacuation in 2000m), and cross-regional vehicle evacuation (20% vehicle evacuation over 2000m). Therefore, these three traffic patterns refer to tsunami shelter accessibility needs and are studied in this paper. They can be measured along with three-hierarchy roads, and include a total of eight accessibility indicators. Each indicator is calculated based on the arithmetic mean of tsunami shelters in each School District sample (see Table 2)..."

[Figure]

**Figure 3: Tsunami evacuation time-space routes with major evacuation activities.**